# MARS: Mamba-driven Adaptive Reordering Scheme for Semantic Occupancy Prediction in Autonomous Driving

## Abstract

Semantic occupancy prediction provides fundamental voxel-level scene perception for autonomous driving systems, yet the massive number of voxels poses significant computational challenges, especially for Transformer-based methods with quadratic complexity. Recently, OccMamba introduces state space models to this task, but its reliance on a handcrafted 3D-to-1D reordering scheme suffers from two critical challenges: (1) *indiscriminate processing of redundant empty voxels*, and (2) *limited adaptivity to diverse scene layouts*. To address this issue, we propose the **M**amba-driven **A**daptive **R**eordering **S**cheme (**MARS**) framework, replacing the static reordering scheme with an adaptive and dynamic design, facilitating modality-aware pruning of redundant empty voxels and scene-adaptive sequence of critical voxels. Specifically, we first introduce the *Adaptive Voxel Pruning (AVP)* module to tackle indiscriminate processing, which filters out redundant empty voxels and retain informative ones, thereby establishing an efficient computational foundation. Then, we present the *Dynamic Voxel Reordering (DVR)* module to address limited adaptivity, which dynamically identifies and sequences critical voxels for scene-level perception, ensuring flexible adaptivity to diverse scenarios. Extensive experiments and analyses on the OpenOccupancy dataset showcase the effectiveness and efficiency of our MARS framework, achieving superior semantic occupancy performance while reducing training memory by 19.6% and accelerating inference by 9.7%.

## 1 Introduction

Semantic occupancy prediction aims to generate a dense voxel-level semantic and geometric representation of the surrounding 3D environment Roldao et al. (2022), providing unified and structured scene comprehension foundation for autonomous driving systems. The pioneering methods, such as JS3C-Net Yan et al. (2021) and MonoScene Cao & de Charette (2022), have made progress with uni-modal partial inputs. Following researches, such as M-CONet Wang et al. (2023b) and FusionOcc Zhang et al. (2024), leverage multi-modal inputs to extract improved visual cues and contextual details, but suffer from the inherent deficiency of CNN He et al. (2016) in modeling long-range global relationships.

To overcome the inherent complexity and scale of autonomous driving scenarios, recent methods Li et al. (2023a); Tong et al. (2023); Zhang et al. (2023); Mei et al. (2024) predominantly adopt Transformer Vaswani et al. (2017) attention architectures to model long-range dependencies within the voxelized scenes. However, the massive number of voxels poses significant challenges to training efficiency due to the quadratic computational complexity of transformer-based networks. In light of this, OccMamba Li et al. (2025) first introduces Mamba Gu & Dao (2023) architecture to the semantic occupancy prediction task, processing 3D-to-1D reordered voxel sequences with linear computation complexity.

Despite its progress, the handcrafted, static 3D-to-1D reordering scheme remains problematic. We conduct a pilot study by varying the proportion of voxels fed into the Mamba blocks in OccMamba, as illustrated in Figure 1 (a). It can be observed that reducing the proportion of voxels to only 1/8 of the full size yields comparable or even slightly higher performance than processing all voxels

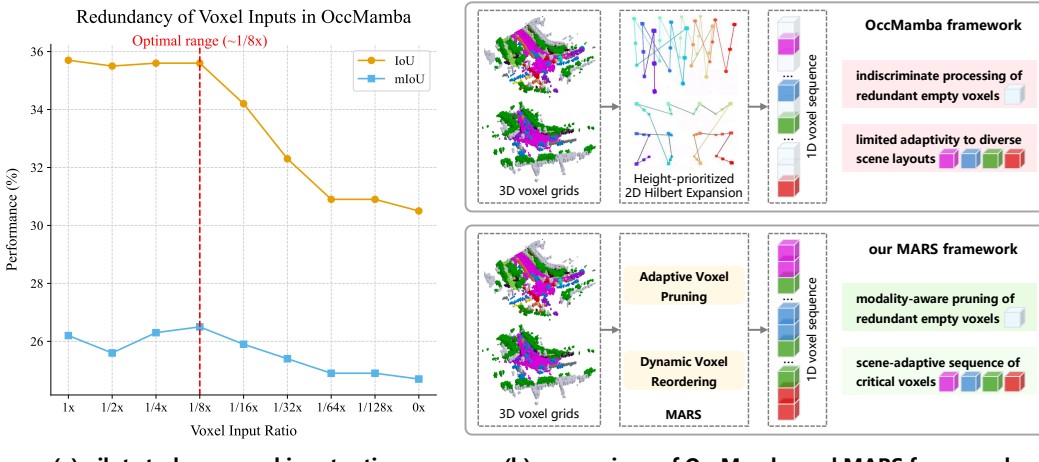

**(a) pilot study on voxel input ratio**  **(b) comparison of OccMamba and MARS framework**

Figure 1: **Motivation for the MARS framework**. **(a)** Our pilot study on OccMamba shows that indiscriminately processing all voxels is suboptimal. Performance is maintained down to a 1/8 input ratio, revealing massive redundancy, yet collapses thereafter, proving the importance of a critical voxel subset. **(b)** This motivates our design, which replaces OccMamba's static scheme with a two-stage adaptive process. Our MARS framework first uses Adaptive Voxel Pruning to tackle redundancy, then employs Dynamic Voxel Reordering to address the lack of adaptivity by creating a context-aware sequence of critical voxels.

throughout the scene, indicating that a large fraction of voxels contributes little to the final prediction and even impairs model performance with redundant computation. In addition, when the voxel input ratio is reduced below 1/16, the performance drops sharply, suggesting the existence of a critical subset of voxels that are essential for maintaining scene-level perception. These findings expose two major limitations of the current 3D-to-1D reordering scheme in OccMamba: (1) **Indiscriminate processing of redundant empty voxels** The handcrafted reordering scheme feeds all voxels into Mamba blocks, overlooking the fact that a large proportion (over 90%) of voxels are empty and causing redundant computation. (2) **Limited adaptivity to diverse scene layouts**. The static reordering scheme fails to adaptively prioritize the critical subset of informative voxels across diver scene layouts, suffering from disruption of non-informative voxels.

To address these issues, we propose the **M**amba-driven **A**daptive **R**eordering **S**cheme (**MARS**) framework, replacing the handcrafted, static reordering scheme with an adaptive and dynamic design, as shown in Figure 1 (b). Specifically, to alleviate the redundancy of processing vast unoccupied regions, we devise the **Adaptive Voxel Pruning** module. Notice that different input modalities exhibit distinct characteristics, emphasizing different regions in the voxel space: LiDAR point clouds provide precise 3D geometric structure information but are inherently sparse, while camera images capture dense semantic cues but lack direct depth information. Therefore, we devise modality adapters to encode modality-aware meta information, followed by a lightweight 3D multi-layer perceptron network identifying informative regions and pruning redundant empty voxels. To overcome the rigidity of handcrafted reordering schemes and adapt to diverse scene layouts, we propose the **Dynamic Voxel Reordering** module. Dynamic driving environments exhibit diverse voxel space distributions, as different scenes consist of distinct arrangements of moving objects, road structures, and free space. In light of this, we design scene adapters that aggregate multi-modal features and exploit contextual voxel dependencies around informative areas, generating dynamic 3D-to-1D sequences of critical voxels with spatial continuity. This dynamic reordering scheme facilitates our MARS framework with modality-aware pruning of redundant empty voxels and scene-adaptive sequence of critical voxels, which flexibly adapts to diverse scene layouts, emphasizing informative critical voxels for state space modeling. Extensive experiments and analyses on the OpenOccupancy Wang et al. (2023b) benchmark demonstrates the effectiveness of our MARS approach.

In summary, our contributions are as follows:

- **An efficient and effective state space model architecture for semantic occupancy prediction**. We present the MARS framework, replacing the handcrafted, static 3D-to-1D reordering scheme with an adaptive and dynamic design, facilitating modality-aware pruning of redundant empty voxels and scene-adaptive sequence of critical voxels for efficient and effective semantic occupancy prediction.

- **An adaptive voxel pruning strategy**. To address the indiscriminate processing of redundant empty voxels, we leverage modality-specific adapters to filter out redundant voxels while preserving informative ones.

- **A dynamic voxel reordering scheme**. To alleviate the limited adaptivity to diverse scene layouts, we exploit contextual voxel dependencies within spatial coherent regions, dynamically prioritizing and sequencing critical voxels across diverse scenarios.

## 2 RELATED WORK

### 2.1 SEMANTIC OCCUPANCY PREDICTION

The primary objective of semantic occupancy prediction is to assess voxel-level occupancy and semantic labels of surrounding scenes. Based on different input modalities, semantic occupancy prediction methods can be broadly divided into three main streams: LiDAR-based methods, camera-based methods, and multi-modal fusion methods.

**LiDAR-based Methods.** Leveraging the natural 3D geometric structural information of LiDAR point clouds, LiDAR-based methods Rist et al. (2021); Cheng et al. (2021); Xia et al. (2023) have long been the predominant solutions to semantic occupancy prediction. UDNet Zou et al. (2021) employs a 3D U-Net architecture to directly construct scene predictions from LiDAR point clouds. LMSCNet Roldao et al. (2020) utilizes 2D convolution networks for lightweight encoding of voxel features. SGCNet Zhang et al. (2018) splits input voxels into distinct groups, enabling sparse spatial group convolutions. JS3C-Net Yan et al. (2021) conducts knowledge fusion between semantic segmentation and occupancy prediction with point-voxel interaction. SSC-RS Mei et al. (2023) designs a multi-branch architecture to integrate semantic and geometric features hierarchically.

**Camera-based Methods.** Due to the cost-effectiveness and flexibility of camera sensors, camera-based methods Miao et al. (2023); Wang & Tong (2024); Wang et al. (2024) have garnered increasing attention with only RGB image inputs. MonoScene Cao & de Charette (2022) first samples image features along lines of sight into voxel representations. Subsequent methods employ Bird's-Eye-View (BEV) representations Li et al. (2022b; 2023c;b) and Tri-Perspective-View (TPV) Huang et al. (2023) representations for more efficient scene modeling. Recently, transformer-based methods Zhang et al. (2023); Wei et al. (2023) adopt transformer architecture with BEV queries Li et al. (2022b), voxel queries Li et al. (2023a), instance queries Jiang et al. (2024), and context queries Yu et al. (2024) to handle the inherent complexity in autonomous driving scenes.

**Multi-modal Fusion Methods.** Towards more accurate details and robust perception, multi-modal fusion methods Gao et al. (2020); Li et al. (2022a); Wang et al. (2023a) leverage different input modalities. AICNet Li et al. (2020) integrates RGB images and depth maps with anisotropic convolution networks. Point-Painting Vora et al. (2020) utilizes image segmentation probabilities to enhance point cloud representations with rich semantics. UniSeg Liu et al. (2023a) makes use of RGB images and three views of LiDAR point cloud for panoptic semantic segmentation. BEVFusion Liu et al. (2023b) unifies multi-modal features in a shared BEV representation space to preserve semantic and geometric information.

However, most mainstream semantic occupancy prediction methods, especially transformer-based ones with quadratic computation complexity, still face significant computational challenges due to the massive number of voxels in large-scale autonomous driving scenarios.

### 2.2 STATE SPACE MODELS

State Space Models (SSMs) Gu et al. (2021); Nguyen et al. (2022) have recently attracted increasing attention as a competitive alternative to transformer Vaswani et al. (2017) architectures for model-

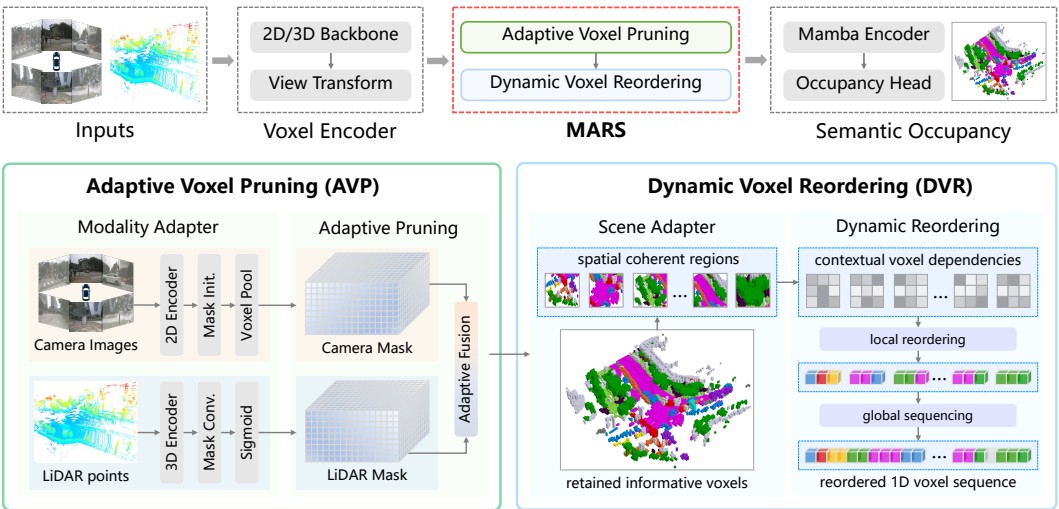

Figure 2: The overall architecture of our MARS approach. The Adaptive Voxel Pruning (AVP) module leverages modality-specific adapters to filter out redundant empty voxels while retaining informative ones. The Dynamic Voxel Reordering (DVR) module exploits contextual voxel dependencies within spatial coherent regions to dynamically prioritize and sequence critical voxels.

ing sequential data with linear computational complexity. Mamba Gu & Dao (2023), notable for processing large-scale data in linear time with selective mechanisms, further extends its variants for processing 2D and 3D data through variants like VMamba Liu et al. (2024), Vision Mamba Zhu et al. (2024), and PointMamba Liang et al. (2024). OccMamba Li et al. (2025) first designs a Mamba-based architecture for semantic occupancy prediction, where 3D voxels are transferred into 1D sequence through a height-prioritized 2D Hilbert reordering scheme. However, the handcrafted, static reordering scheme suffers from redundant processing of empty voxels and limited adaptivity to dynamic scenarios. To address this issue, we propose MARS, an adaptive and dynamic 3D-to-1D reordering scheme for improved computational efficiency and scene adaptivity.

## 3 METHODOLOGY

In this section, we first briefly review state space models (SSMs) and its application in semantic occupancy prediction (OccMamba). Then, we introduce the *Mamba-based Adaptive Reordering Scheme (MARS)* framework, an efficient state space model architecture that leverages adaptive and dynamic 3D-to-1D reordering scheme for improved computational efficiency and scene adaptivity. As illustrated in Figure 2, the overall framework of our MARS approach consists of two key components: (1) an Adaptive Voxel Pruning (AVP) module that leverages modality-specific adapters to filter out redundant voxels while preserving informative ones; (2) a Dynamic Voxel Reordering (DVR) module that exploits contextual voxel dependencies to dynamically prioritize and sequence critical voxels.

### 3.1 PRELIMINARIES

**Problem Setup.** The primary objective of semantic occupancy prediction is to generate a dense representation of the surrounding environment, assigning both occupancy states and semantic classes to a predefined 3D voxel grids. Formally, given a set of multi-view camera images $\mathcal{I} = \{I_i\}_{i=1}^{N_{cam}}$ and LiDAR point clouds $\mathcal{P}$, the task is to predict a voxel grid $\mathcal{V} \in \mathbb{R}^{X \times Y \times Z}$, where $X, Y, Z$ correspond to the grid's resolution. Each voxel is assigned a semantic label of either empty denoted by $c_0$ or occupied by one of the predefined semantic classes in $C \in \{c_1, \cdots, c_N\}$.

**State space models.** State space models (SSMs) are inspired by the control theory and formulates sequence modeling as a structured state transition process, where the hidden state evolves according

to linear dynamical systems and is updated with input-dependent parameters. This process can be formulated as:

$$h_t = A h_{t-1} + B x_t, \quad y_t = C h_t + D x_t \tag{1}$$

where $h_t$ is the hidden state, $x_t$ is the input, and $y_t$ is the output. Recent variants such as Mamba Gu & Dao (2023) enhance SSMs with selective state update and efficient parallelization, modeling long-range dependencies with linear computations complexity.

**OccMamba.** OccMamba Li et al. (2025) first introduces the Mamba-based architectures into semantic occupancy prediction, consisting of multi-modal visual encoders, an OccMamba encoder, and an occupancy head. It converts 3D voxelized grids into 1D voxel sequences through a height-prioritized 2D Hilbert reordering scheme, then employs hierarchical mamba modules and local context processors to model scene-level long-range dependencies. However, the handcrafted and static 3D-to-1D reordering scheme suffers from issues of indiscriminate processing of redundant empty voxels and limited adaptivity to diverse scene layouts, thus limiting overall performance of semantic occupancy prediction.

## 3.2 ADAPTIVE VOXEL PRUNING

To address the issue of indiscriminate processing of redundant empty voxels, we propose the Adaptive Voxel Pruning (AVP) module, leveraging modality-aware information to filter out redundant voxels while retaining informative regions. The key insight behind AVP is that considering the large proportion of empty voxels (over 90%) and complementary characteristics of different input modalities, it is crucial to leverage modality-aware metadata together with extracted features to adaptively identify informative voxels, pruning redundant ones for improved computational efficiency.

**Modality Adapter.** To fully exploit the inherent characteristics of different modalities, the modality adapter leverages modality-aware metadata to generate heuristic geometric structures within the voxel space, serving as spatial priors for redundant voxel pruning.

For camera image inputs, we leverage camera parameters and depth probabilities as metadata to compensate for the lack of explicit 3D structural information. Specifically, we first initialize the heuristic image mask $M_{\text{heu}} \in \mathbb{R}^{1 \times H_{\text{img}} \times W_{\text{img}}}$ with all elements set to 1, indicating the potential contribution of each pixel. Then, the DepthNet CS Kumar et al. (2018) is utilized to generate pixel-wise depth probabilities $D \in \mathbb{R}^{D \times H_{\text{img}} \times W_{\text{img}}}$, lifting image masks along the depth dimension. Finally, the lifted masks are regularized into voxelized grid coordinates through voxel pooling Philion & Fidler (2020) with camera parameters. The above process can be formulated as follows:

$$M_{\text{cam}} = \text{VoxelPool}\left((M_{\text{heu}} \cdot D), K, T\right) \tag{2}$$

where $K, T$ are the camera intrinsic and extrinsic matrices, and the voxel pooling operation accumulates depth-aware contributions of each pixel, generating camera voxel mask $M_{\text{cam}} \in \mathbb{R}^{X \times Y \times Z}$.

For LiDAR point cloud inputs, they naturally encode explicit 3D structural information with 3D coordinates and reflection intensity, which is well-suited for extracting spatial priors. To fully exploit these properties, we first regularize the input point clouds $P$ into voxelized grids Zhou & Tuzel (2018), where each voxel aggregates both geometric coordinates and intensity values of points falling inside it. The voxelized features $V_{\text{lidar}} \in \mathbb{R}^{C \times X \times Y \times Z}$ are then fed into a lightweight multi-layer perceptron $\psi_{\text{lidar}}(\cdot)$ to capture geometry-aware spatial contexts:

$$M_{\text{lidar}} = \sigma\left(\psi_{\text{lidar}}(V_{\text{lidar}})\right) \tag{3}$$

where $\sigma(\cdot)$ is the sigmoid function, and $M_{\text{lidar}} \in \mathbb{R}^{X \times Y \times Z}$ is the LiDAR voxel mask, providing spatial priors within the voxel space based on LiDAR structural and intensity cues.

**Adaptive Pruning.** Given the modality-aware voxel masks $M_{\text{cam}}, M_{\text{lidar}}$, we further design an adaptive pruning module to fuse multi-modal features and filter out redundant voxels. Camera masks highlight regions aligned with projected image features, while LiDAR masks emphasize structural priors in occupied space. We generate adaptive fusion weights $W \in \mathbb{R}^{X \times Y \times Z \times 2}$ from the concatenation of multi-modal voxel features and masks $[V_{\text{cam}}, M_{\text{cam}}; V_{\text{lidar}}, M_{\text{lidar}}]$ using 3D convolution

with softmax on the last dimension. The fused voxel features and masks are computed as follows:

$$V = W_0 \odot M_{\text{cam}} \odot V_{\text{cam}} + W_1 \odot M_{\text{lidar}} \odot V_{\text{lidar}}$$
$$M = W_0 \odot M_{\text{cam}} + W_1 \odot M_{\text{lidar}} \tag{4}$$

where $\odot$ is the element-wise multiplication, $V$ is the fused voxel features, and $M$ is the adaptive voxel mask with voxel-wise occupancy scores based on multi-modal spatial priors, facilitating redundant voxel pruning with confidence threshold $\mathbf{1}_{M>\theta}$.

### 3.3 Dynamic Voxel Reordering

To address the issue of limited adaptivity to diverse scene layouts, we propose the Dynamic Voxel Reordering (DVR) module, exploiting contextual voxel dependencies to dynamically prioritize and sequence critical voxels across diverse scenarios. The design rationale of DVR is that since diverse driving environments consist of distinct voxel space distributions, it is essential to exploit scene-aware voxel dependencies and dynamically reorder critical voxels, enhancing the adaptivity to diverse scenarios. DVR takes the fused features $V$ and pruning mask $M$ from AVP as input.

**Scene Adapter.** To generate a reordering scheme that is aware of the overall scene layout, the scene adapter is designed to discover and summarize spatially coherent regions of interest. We first employ a connected components labeling algorithm on the pruning mask to partition the set of informative voxels $\mathbf{1}_{M>\theta}$ into $K$ disjoint salient regions $\{M_1, M_2, \cdots, M_K\}$. Then for each identified region $M_k$, we generate a representative context vector by applying an aggregation function over its constituent voxels. Specifically, we use average pooling for global aggregation:

$$m_k = \text{AvgPool}(V[M_k]) \tag{5}$$

where $m_k$ is the representative vector of informative voxels within region $M_k$. This process yields a set of regional context vectors , each summarizing a semantically and spatially cohesive part of the scene, thereby providing richer and more structured spatial priors for dynamic reordering.

**Dynamic Reordering.** With the partition of spatially coherent regions and the extraction of regional representative vectors, the dynamic reordering process generates a scene-adaptive 1D voxel sequence, where the importance score of each voxel is conditioned on the coherent regions it belongs to. Specifically, for each voxel $v_i$ located in region $M_k$, we concatenate its feature with the corresponding regional representative vector. The combined representations are then fed into a shared MLP, $\phi_{\text{order}}$, predicting voxel-wise reordering score $s_i$:

$$s_i = \phi_{\text{order}}\left(\text{Concat}(v_i, m_k)\right), \quad \text{where } i \in M_k \tag{6}$$

Voxels within each spatial coherent region are locally reordered based on their reordering scores. Then, the reordered sequences are globally concatenated together and fed into the Mamba encoder blocks. By conditioning the importance score on regional context, this scheme encourages the model to group and prioritize related information. For example, all voxels belonging to a "pedestrian" cluster might be ranked closely together. This enhances the locality of the input sequence and allows the state space model to build a more coherent understanding of distinct scene elements, leading to superior adaptivity across diverse scenarios.

### 3.4 Training Objective.

Following Li et al. (2025), we adopt the cross-entropy loss $\mathcal{L}_{\text{ce}}$ for classification, the lovasz-softmax loss $\mathcal{L}_{\text{lovasz}}$ Berman et al. (2018) for semantic segmentation, the scene-class affinity loss $\mathcal{L}_{\text{scal}}$ Cao & de Charette (2022) for spatial alignments, and the depth supervision loss $\mathcal{L}_{\text{depth}}$ Li et al. (2023c) for depth estimation. The final training objective is formulated as follows:

$$\mathcal{L} = \mathcal{L}_{\text{ce}} + \mathcal{L}_{\text{lovasz}} + \mathcal{L}_{\text{scal}} + \mathcal{L}_{\text{depth}} \tag{7}$$

## 4 Experiments

### 4.1 Experimental Setup

**Datasets.** We evaluate our MARS approach on the OpenOccupancy Wang et al. (2023b) dataset. OpenOccupancy extends the popular nuScenes Caesar et al. (2020) dataset with dense semantic

Table 1: Quantitative comparisons on the OpenOccupancy Wang et al. (2023b) validation set with v0.0 annotations. C, D, L denote camera, depth and LiDAR, respectively. Best results are highlighted in **bold**, and second-best results are underlined.

| Method | Input Modality | IoU | mIoU | barrier | bicycle | bus | car | const. veh. | motorcycle | pedestrian | traffic cone | trailer | truck | drive surf. | other flat | sidewalk | terrain | manmade | vegetation |
|---|---|---|---|---|---|---|---|---|---|---|---|---|---|---|---|---|---|---|---|
| MonoScene 2022 | C | 18.4 | 6.9 | 7.1 | 3.9 | 9.3 | 7.2 | 5.6 | 3.0 | 5.9 | 4.4 | 4.9 | 4.2 | 14.9 | 6.3 | 7.9 | 7.4 | 10.0 | 7.6 |
| TPVFormer 2023 | C | 15.3 | 7.8 | 9.3 | 4.1 | 11.3 | 10.1 | 5.2 | 4.3 | 5.9 | 5.3 | 6.8 | 6.5 | 13.6 | 9.0 | 8.3 | 8.0 | 9.2 | 8.2 |
| SparseOcc 2024 | C | 21.8 | 14.1 | 16.1 | 9.3 | 15.1 | 18.6 | 7.3 | 9.4 | 11.2 | 9.4 | 7.2 | 13.0 | 31.8 | 21.7 | 20.7 | 18.8 | 6.1 | 10.6 |
| 3DSketch 2020 | C&D | 25.6 | 10.7 | 12.0 | 5.1 | 10.7 | 12.4 | 6.5 | 4.0 | 5.0 | 6.3 | 8.0 | 7.2 | 21.8 | 14.8 | 13.0 | 11.8 | 12.0 | 21.2 |
| AICNet 2020 | C&D | 23.8 | 10.6 | 11.5 | 4.0 | 11.8 | 12.3 | 5.1 | 3.8 | 6.2 | 6.0 | 8.2 | 7.5 | 24.1 | 13.0 | 12.8 | 11.5 | 11.6 | 20.2 |
| LMSCNet 2020 | L | 27.3 | 11.5 | 12.4 | 4.2 | 12.8 | 12.1 | 6.2 | 4.7 | 6.2 | 6.3 | 8.8 | 7.2 | 24.2 | 12.3 | 16.6 | 14.1 | 13.9 | 22.2 |
| JS3C-Net 2021 | L | 30.2 | 12.5 | 14.2 | 3.4 | 13.6 | 12.0 | 7.2 | 4.3 | 7.3 | 6.8 | 9.2 | 9.1 | 27.9 | 15.3 | 14.9 | 16.2 | 14.0 | 24.9 |
| M-CONet 2023b | C&L | 29.5 | 20.1 | 23.3 | 13.3 | 21.2 | 24.3 | 15.3 | 15.9 | 18.0 | 13.3 | 15.3 | 20.7 | 33.2 | 21.0 | 22.5 | 21.5 | 19.6 | 23.2 |
| Co-Occ 2024 | C&L | 30.6 | 21.9 | 26.5 | 16.8 | 22.3 | 27.0 | 10.1 | 20.9 | 20.7 | 14.5 | 16.4 | 21.6 | 36.9 | 23.5 | 5.5 | 23.7 | 20.5 | 23.5 |
| OccMamba 2025 | C&L | **35.7** | 26.2 | 30.2 | 20.5 | **26.5** | 29.5 | **18.8** | 26.0 | 23.7 | 19.9 | **20.6** | 25.4 | 38.4 | 26.5 | 27.0 | 26.6 | 28.9 | 30.5 |
| **MARS (ours)** | C&L | **36.2** | **27.1** | **31.8** | **23.0** | 26.3 | **31.0** | 17.4 | **27.8** | **27.8** | **20.1** | 20.4 | **26.1** | **39.5** | **27.2** | **27.7** | **27.1** | **29.0** | **30.9** |

Table 2: Efficiency evaluation results against SSM-based method OccMamba Li et al. (2025) and Transformer-based method M-CONet Wang et al. (2023b) on the OpenOccupancy validation set, taking both camera images and LiDAR point clouds as inputs.

| Method | Training Memory (GB) | Inference Time (ms) | Input Voxel | mIoU (%) |
|---|---|---|---|---|
| M-CONet | 37.3 | 2,231 | - | 20.1 |
| OccMamba-384 | 37.7 | 2,401 | 163,840 | 26.2 |
| OccMamba-128 | 23.1 | 2,027 | 163,840 | 25.2 |
| MARS-384 | 30.3 | 2,168 | $\sim$20,480 | 27.1 |
| MARS-128 | 22.3 | 1,895 | $\sim$20,480 | 25.7 |

occupancy annotations, which comprises 700 training sequences and 150 validation sequences, with a total annotation of 17 semantic classes. The semantic occupancy labels are represented within $512 \times 512 \times 40$ voxelized grids, with a voxel resolution of $0.2m$.

**Evaluation Metrics.** Following Li et al. (2025), we employ the Intersection over Union (IoU) of occupied voxels as the evaluation metric for the task of class-agnostic scene scene (SC). Additionally, we report the mean Intersection over Union (mIoU) across all semantic categories to measure the performance of the semantic scene completion (SSC):

$$\text{IoU} = \frac{TP}{TP + FP + FN}, \quad \text{mIoU} = \frac{1}{C} \sum_{c=1}^{C} \frac{TP_c}{TP_c + FP_c + FN_c} \tag{8}$$

where $TP, FP, FN$ represent the number of true positive, false positive, and false negative occupancy predictions, and $C$ stands for the total number of classes.

**Implementation Details.** We employ the ResNet-50 He et al. (2016) network as the image backbone. The Mamba Encoder and Decoder maintains the architecture in OccMamba Li et al. (2025), which contains four groups and each group consists of two Mamba blocks. For the OpenOccupancy dataset, we follow the official input settings Wang et al. (2023b), where six surround view images are used as camera inputs, together with a fusion of ten frames of LiDAR points covering the range of $[-51.2m \sim 51.2m, -51.2m \sim 51.2m, -2.0m \sim 6.0m]$. The feature dimension within the Mamba blocks is set to 384. The threshold $\theta$ for adaptive pruning is set default to 0.5. We train MARS for 20 epochs on 8 NVIDIA A6000 GPUs, with a total batch size of 8. The AdamW Loshchilov & Hutter (2017) optimizer is adopted with an initial learning rate of 5e-4 and a weight decay of 1e-2.

Table 3: Ablation study on the OpenOccupancy validation set, investigating the effectiveness of different architectural components of our MARS approach with different input modalities. C denotes camera inputs and L denotes LiDAR inputs, respectively.

| Variants | | C | | L | | C&L | |
|---|---|---|---|---|---|---|---|
| AVP | DVR | IoU | mIoU | IoU | mIoU | IoU | mIoU |
| | | 19.3 | 12.1 | 35.4 | 22.7 | 35.7 | 26.2 |
| ✓ | | 19.8 | 12.6 | 35.7 | 23.0 | 35.9 | 26.3 |
| ✓ | ✓ | 20.4 | 13.0 | 36.5 | 23.6 | 36.2 | 27.1 |

## 4.2 MAIN RESULTS

As shown in Table 1, we compare MARS with existing state-of-the-art methods on the OpenOccupancy validation set. It can be observed that our proposed MARS approach achieves a superior performance of **36.2%** IoU and **27.1%** mIoU, demonstrating its effectiveness over the strong OccMamba baseline. It is crucial to highlight that MARS achieves this result while operating on a dynamically pruned subset of voxels for improved computational efficiency. Table 2 presents the efficiency evaluations results of our MARS against SSM-based method OccMamba and Transformer-based method M-CONet, where the networks are trained on 8 NVIDIA A6000 GPUs and perform inference on a single NVIDIA A6000 GPU. Compared to OccMamba's processing all $128 \times 128 \times 10 = 163,840$ voxels indiscriminately, our MARS feeds only $\sim$**12.5%** of the total voxels into the Mamba blocks, reducing the training memory by **19.6%** (from 37.7GB to 30.3GB) and accelerating inference time by **9.7%** (from 2,401ms to 2,168ms). The above concurrent improvements in both effectiveness and efficiency are the direct outcome of our targeted architectural design, which systematically addresses the two key issues identified in our pilot study. Specifically, the Adaptive Voxel Pruning (AVP) module directly confronts the problem of indiscriminate processing. By intelligently filtering out vast, non-informative regions, it establishes a computationally efficient foundation, ensuring that the model's resources are not wasted on empty space. Building upon this sparse yet salient representation, the Dynamic Voxel Reordering (DVR) module then tackles the challenge of limited adaptivity. It analyzes the contextual relationships between the retained voxels and sequences them into a coherent, scene-aware stream that is optimal for state space modeling. This synergy—where AVP first provides the voxel efficiency and DVR then unlocks the performance from that critical information—validates our adaptive approach.

## 4.3 ABLATION STUDY

To further investigate the effectiveness of our MARS approach and different components, we conduct ablation experiments on the OpenOccupancy validation set as follows:

**Ablation on Network Components.** As shown in Table 3, we take OccMamba Li et al. (2025) as the baseline method and present the results of ablation experiments with different modality inputs. We take OccMamba Li et al. (2025) as our baseline, where a handcrafted, static height-prioritized 2D Hilbert expansion is utilized to process all voxels with Mamba blocks. By integrating the Adaptive Voxel Pruning (AVP) module, we effectively filter out a majority of the redundant, empty voxels while retaining the informative ones. AVP provides the voxel foundation for further efficiency and effectiveness gains, and still achieves competitive performance with different modality inputs. This underscores AVP's ability to adaptively preserve informative voxels for scene understanding. Subsequently, we introduce the Dynamic Voxel Reordering (DVR) module on top of AVP to dynamically prioritize and sequence critical voxels by exploits contextual voxel dependencies, further improves the semantic occupancy prediction performance with 1.1% IoU and 0.9% mIoU improvements for camera inputs, 1.1% IoU and 0.9% mIoU improvements for LiDAR inputs, and 0.5% IoU and 0.9% mIoU improvements for multi-modal inputs, respectively. These improvements validate that the synergy of first pruning redundancy (AVP) and then intelligently structuring the remaining information (DVR) is key to our framework's success.

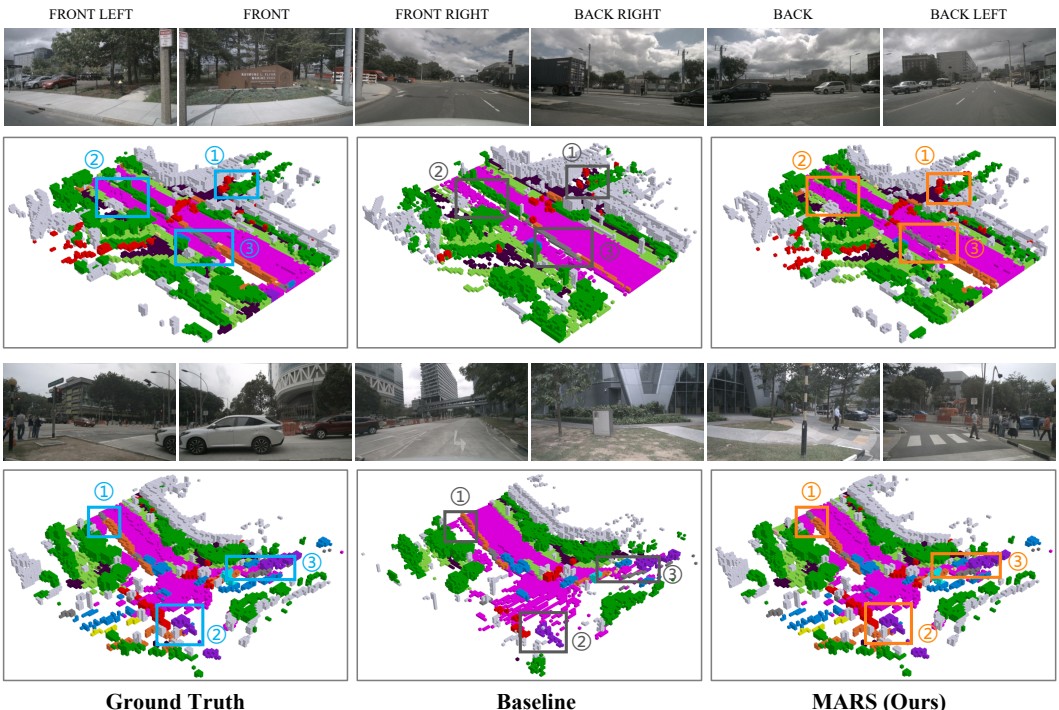

Figure 3: Visualization results on the OpenOccupancy validation set. The occupancy ground truth is outlined with blue boxes. While black boxes indicate erroneous occupancy predictions of the baseline method, and orange boxes highlight more accurate predictions by our MARS approach. Better viewed when zoomed in.

## 4.4 VISUALIZATIONS

Figure 3 demonstrates the visualization results from the OpenOccupancy validation set. The surround-view input images are illustrated in the first and third lines. In the first row, the occupancy ground truth is outlined with blue boxes. The second row presents the occupancy predictions generated by the baseline method, where false predictions are indicated with black boxes. While the third row displays the results of our MARS approach, and orange boxes highlight our refinement for more accurate occupancy predictions. It can be observed that the handcrafted, static reordering scheme of OccMamba struggles to maintain the spatial continuity of large entities, while indiscriminate processing of redundant voxels introduces confusing semantics. In contrast, our MARS generates more complete and accurate semantic occupancy predictions, demonstrating the effectiveness of AVP's adaptively filtering out redundant empty voxels and DVR's dynamically sequencing critical voxels for improved semantic occupancy prediction performance.

## 5 CONCLUSION

In this work, we addressed the critical limitations inherent in the static reordering schemes used by state space models for semantic occupancy prediction: the indiscriminate processing of redundant voxels and limited adaptivity to diverse scene layouts. To address these issues, we introduce MARS, a Mamba-driven Adaptive Reordering Scheme, replacing this static design with an adaptive and dynamic approach. MARS is composed of two synergistic modules: the Adaptive Voxel Pruning (AVP) module leverages multi-modal priors to adaptively prune redundant empty voxels, and the Dynamic Voxel Reordering (DVR) module analyzes the remaining informative voxels to generate a scene-aware, contextually prioritized 1D voxel sequence. Extensive experiments on the OpenOccupancy benchmark demonstrate that MARS achieves superior semantic occupancy prediction performance, while delivering substantial improvements in computational efficiency, significantly reducing both training memory and inference latency compared to the baseline.

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
