# OpenReview forum: "MARS: Mamba-driven Adaptive Reordering Scheme for Semantic Occupancy Prediction in Autonomous Driving"
_ICLR.cc/2026/Conference — Submitted to ICLR 2026_

### Official Review · Reviewer_UoGh · 2025-10-26

**Soundness:** 3
**Presentation:** 3
**Contribution:** 2
**Rating:** 4
**Confidence:** 5

**Summary:**

This paper addresses OccMamba’s limitations of the indiscriminate processing of redundant empty voxels and limited adaptivity to diverse scene layouts. To solve these issues, two core modules, the Adaptive Voxel Pruning (AVP) module and the Dynamic Voxel Reordering (DVR) module are proposed. The AVP module leverages modality-aware metadata to filter out redundant empty voxels while retaining informative ones. The DVR module then exploits contextual dependencies within spatially coherent regions to dynamically prioritize and sequence critical voxels. Extensive experiments on the OpenOccupancy dataset demonstrate that MARS achieves superior performance compared to baseline methods like OccMamba. Ablation studies further validate the synergistic effectiveness of the AVP and DVR modules.

**Strengths:**

1. The paper identifies two key limitations of OccMamba via a pilot study shown in Fig 1. This pilot study provides empirical support for the necessity of voxel pruning and dynamic reordering.
2. MARS is compared with SOTA methods such as OccMamba and M-CONet, achieves comparable efficiency and accuracy performance.
3. The details of AVP and DVR are clearly presented, which facilitates the reproduction of subsequent methods.

**Weaknesses:**

1. Both the AVP and DVR have limited innovation. AVP is a commonly utilized optimization for voxel-based methods. DVR serves a role similar to self-attention and has also been widely adopted in other methods.
2. Experiments are only on OpenOccupancy, while OccMamba also conducts experiments on SemanticKITTI. This casts doubt on the model's generalization ability.
3. Although MARS has improved the efficiency of OccMamba, its latency of over 2s is still far from practical requirements. The authors should evaluate the accuracy and efficiency of a small-scale model.

**Questions:**

1. What do “384/128” denote in Table 2? Is it possible to try more values to observe their impact on efficiency?
2. In Figure 1 (a), why is there a sudden drop in mIoU when 1/2 of the voxels are used?

---

### Official Review · Reviewer_z5Dk · 2025-10-29

**Soundness:** 2
**Presentation:** 3
**Contribution:** 1
**Rating:** 2
**Confidence:** 4

**Summary:**

This paper proposes MARS for semantic occupancy prediction in autonomous driving. MARS replaces the handcrafted 3D-to-1D reordering scheme in OccMamba with two new modules: AVP and DVR. The authors demonstrate that MARS feeds only about 12.5% of total voxels into Mamba blocks. AVP and DVR is verified by ablation analyses.

**Strengths:**

The paper identifies two primary issues in OccMamba—computational redundancy and limited adaptivity—and supports the claim through a study employing a 1/8 voxel ratio experiment, which is both clear and well-motivated. The writing is clear and easy to follow. The experimental results demonstrate improvements in both efficiency (−19.6% training memory) and accuracy (+0.9 mIoU) on a single dataset.

**Weaknesses:**

The paper represents an incremental extension of OccMamba, offering limited technical and theoretical novelty. The experimental evaluation is confined to a single dataset (OpenOccupancy). DVR’s learned ordering lacks a thorough analysis of its influence on Mamba’s state-space modeling.

**Questions:**

1. The evaluation is conducted solely on OpenOccupancy. The authors should also evaluate on SemanticKITTI and SemanticPOSS.
2. In the AVP module, how does the model determine the occupancy of voxels that are occluded in the camera view and absent from sparse LiDAR returns? How can a 3D convolution correctly infer their occupancy states in such cases?
3. The adaptive aspect of AVP remains unclear. Is it implemented via multimodal feature fusion followed by a confidence threshold?
4. Please provide an analysis of how the ordering of critical voxels in DVR affects the state-space model’s sequence modeling. The accuracy improvement shown in Tab. 1 appears modest.
5. Why does the training objective differ from that of OccMamba? Is this change essential for the reported performance?
6. Given that 3D-to-1D mapping inherently breaks spatial adjacency, does the proposed dynamic reordering further disrupt spatial coherence? How does the model mitigate this effect?

---

### Official Review · Reviewer_aCX2 · 2025-11-02

**Soundness:** 3
**Presentation:** 3
**Contribution:** 2
**Rating:** 4
**Confidence:** 4

**Summary:**

This paper proposes a mamba-driven framework with adaptive voxel pruning and dynamic voxel reordering to improve the efficiency and generalization for multimodal 3D occupancy prediction. The experimental results demonstrate the effectiveness of the proposed method.

**Strengths:**

1. The paper is well-written and easy to follow.
2. The proposed framework reduces computational cost while maintaining or improving accuracy compared to baseline approaches.

**Weaknesses:**

1. The in-text citation format is incorrect. For example, "OccMamba Li et al. (2025)" should be "OccMamba (Li et al., 2025)", "Mamba Gu & Dao (2023)" should be "Mamba (Gu & Dao, 2023)", etc.
2. Lack of important sensitivity analysis of confidence threshold \theta.
3. Is there any supervision for voxel-wise reordering scores s_i? If not, are the reordered sequences multiplied by the score? Otherwise, there is no gradient backpropagated from the final loss to the score, and therefore, the score is not reliable.
4. The improvement in performance or efficiency is marginal compared to the baseline approach. Some previous methods, such as SDGOCC, can achieve high speed while maintaining a good performance.
5. Lack of results on another commonly used dataset, Occ3D-nuScenes. The main results are solely reported on OpenOccupancy—there are no comparisons with many recent papers on the common Occ3D-nuScenes dataset, such as SDGOCC and RIOcc. Specifically, SDGOCC focuses on real-time 3D occupancy prediction, while RIOcc is a memory-efficient approach.
6. The proposed method is optimized upon a mamba-based baseline framework, without clear evidence to generalize to other approaches such as convolution- or transformer-based ones.

SDGOCC: Semantic and Depth-Guided Bird's-Eye View Transformation for 3D Multimodal Occupancy Prediction. CVPR 2025.

RIOcc: Efficient Cross-Modal Fusion Transformer with Collaborative Feature Refinement for 3D Semantic Occupancy Prediction. ICCV 2025.

**Questions:**

1. It is not clear whether the heuristic image mask M_{heu} is either a learnable mask initialized to ones or a predicted mask by the network.
2. Is there any supervision for the image mask and lidar mask with ground-truth mask labels?

---

### Meta-Review · Area_Chair_NxFf · 2026-01-06

**Summary:**

The reviewers raised several key concerns that inform the decision for this paper. Reviewer aCX2 highlighted issues with citation format, lack of sensitivity analysis for the confidence threshold, unclear supervision for voxel reordering scores, marginal improvements over baselines, absence of results on Occ3D-nuScenes, and limited generalization beyond Mamba-based approaches. Reviewer z5Dk criticized the incremental novelty, evaluation confined to OpenOccupancy without SemanticKITTI or SemanticPOSS, unresolved questions about handling occlusions in the AVP module, unclear adaptivity in AVP, lack of analysis on DVR's ordering impact, unexplained changes in training objective, and potential disruption of spatial coherence. Reviewer UoGh noted limited innovation in AVP and DVR, insufficient generalization due to single-dataset evaluation, and high latency that remains impractical, suggesting a need for small-scale model tests.

**Reviewer Concerns:**

Based on the reviews, many issues likely remain outstanding. Concerns such as citation errors (aCX2) could be easily fixed, but fundamental issues like lack of multi-dataset evaluation (aCX2, z5Dk, UoGh), insufficient novelty (z5Dk, UoGh), and technical gaps in AVP/DVR explanations (e.g., supervision for reordering scores, occlusion handling, and spatial coherence) appear unaddressed and critical.

**Reviewer Scores:**

No change in the scores.
Reviewer aCX2 (4),
Reviewer z5Dk (2),
Reviewer UoGh (4).

---

### Decision · Program_Chairs · 2026-01-26

Reject